# Investigating Optimal Time Step Intervals of Imaging for Data Quality through a Novel Fully-Automated Cell Tracking Approach

**DOI:** 10.3390/jimaging6070066

**Published:** 2020-07-07

**Authors:** Feng Wei Yang, Lea Tomášová, Zeno v. Guttenberg, Ke Chen, Anotida Madzvamuse

**Affiliations:** 1Department of Chemical and Process Engineering, University of Surrey, Stag Hill, University Campus, Guildford GU2 7XH, UK; 2Ibidi GmbH Lochhammer Schlag 11, 82166 Gräfelfing, Germany; ltomasova@ibidi.de (L.T.); zguttenberg@ibidi.de (Z.v.G.); 3Department of Mathematical Sciences, University of Liverpool, Liverpool L69 7ZL, UK; k.chen@liverpool.ac.uk; 4School of Mathematical and Physical Sciences, Department of Mathematics, University of Sussex, Brighton BN1 9QH, UK; 5Department of Mathematics, University of Johannesburg, P.O. Box 524, Auckland Park 2006, Johannesburg, South Africa

**Keywords:** optimal time step intervals, microscope data acquisition, fully-automated cell tracking, phase-contrast microscopy, segmentation, particle tracking, chemotaxis, directed cell migration, tracking accuracy

## Abstract

Computer-based fully-automated cell tracking is becoming increasingly important in cell biology, since it provides unrivalled capacity and efficiency for the analysis of large datasets. However, automatic cell tracking’s lack of superior pattern recognition and error-handling capability compared to its human manual tracking counterpart inspired decades-long research. Enormous efforts have been made in developing advanced cell tracking packages and software algorithms. Typical research in this field focuses on dealing with existing data and finding a best solution. Here, we investigate a novel approach where the quality of data acquisition could help improve the accuracy of cell tracking algorithms and vice-versa. Generally speaking, when tracking cell movement, the more frequent the images are taken, the more accurate cells are tracked and, yet, issues such as damage to cells due to light intensity, overheating in equipment, as well as the size of the data prevent a constant data streaming. Hence, a trade-off between the frequency at which data images are collected and the accuracy of the cell tracking algorithms needs to be studied. In this paper, we look at the effects of different choices of the time step interval (i.e., the frequency of data acquisition) within the microscope to our existing cell tracking algorithms. We generate several experimental data sets where the true outcomes are known (i.e., the direction of cell migration) by either using an effective chemoattractant or employing no-chemoattractant. We specify a relatively short time step interval (i.e., 30 s) between pictures that are taken at the data generational stage, so that, later on, we may choose some portion of the images to produce datasets with different time step intervals, such as 1 min, 2 min, and so on. We evaluate the accuracy of our cell tracking algorithms to illustrate the effects of these different time step intervals. We establish that there exist certain relationships between the tracking accuracy and the time step interval associated with experimental microscope data acquisition. We perform fully-automatic adaptive cell tracking on multiple datasets, to identify optimal time step intervals for data acquisition, while at the same time demonstrating the performance of the computer cell tracking algorithms.

## 1. Introduction

Cell migration is a fundamental process that is essential to life and it is linked to many important physiological and pathological events such as immune response, wound healing, tissue differentiation, embryogenesis, and tumour invasion [1,2,3,4,5,6,7]. Modern microscopy with image capturing devices is at the heart of techniques to study live-cell migration, such as those driven by chemotactic signalling during directed migration [8,9,10,11]. The advent of high-throughput, high-resolution microscopy, and imaging techniques means that cell outlines and migration trajectories or pathways may be visually accessible and manually traceable by a human operator (through manual tracking) to result in what is known as the gold-standard ground truth.

With the ever-increasing use of microscopy, a humongous amount of experimental data is routinely generated in many laboratories. Most of these data are collected in the absence of a monitoring segmentation and tracking algorithm. Due to the sheer magnitude of the data involved, manual tracking is often cumbersome and the development of computer algorithms for automated cell tracking is, thus, highly desirable [12,13,14,15]. In contrast, a noticeable downside of fully-automated tracking is that the recognition capability of computers is very limited (comparing to a trained human operator) despite the sizeable efforts made from the wider academic community [16,17,18,19,20].

Most of the research focuses on building fully-automated algorithms that post-process experimental outcomes without investigating the procedures underpinning data generation. The algorithms are thus blind to the underlying properties associated with quality data acquisition and a key parameter is the time step interval within the microscope. We have not explored any other parameters, since we feel these do not correlate directly with the cell tracking algorithm. Studies of this nature could be undertaken in a similar fashion. It is here that the novelty of our contribution lies, we strongly believe (and will demonstrate) that the time step interval embedded within the microscope camera setting at which experimental images are recorded is a crucial parameter that should influence the accuracy of the computer algorithms more drastically. The common choice of the time step interval between consecutive and successive camera shoot images currently published in the literature are in an ad-hoc trial-and-error approach that is generally based on the intuition and experience of microscopist or experimentalist. Typically, 5 and 10 min time step intervals are among the most common choices, for example, [21,22].

By using a novel approach for fully-automated algorithms for cell tracking, we investigate viable options during experimental microscope data acquisition to allow for accurate and high-quality acquisition of images, thereby contributing to fully-automatic big data analytics in cellular biology. Here, we focus on different choices of the time step interval within the microscope. Taking into account the drawbacks that arise from using computer algorithms (i.e., lack of superior pattern recognition and error-handling capability compared to its human counterpart), one may propose to take constant and frequent sequences of images with an exceptionally short time step interval (e.g., a few seconds apart or even multiple frames within a second). However, when an image is taken, light is shed on the cells for clear visualisation. Yet, large doses of light have the potential to damage or even kill the cells under experiment [23,24,25,26]. Recent studies show that yeast cells that appeared healthy directly after irradiation with a very low light-dose from a fluorescent microscope failed to divide when left overnight, whereas their non-imaged neighbours divided normally [27]. In fact, it has been pointed out that fixation, plasma membrane permeabilisation, and cytoskeleton destruction are among the typical light-induced cell damages in live-cell microscopy [24,28].

Evidently, a trade-off emerges on the choice of the time step interval and the quality of the images collected. Images acquired with a short time step interval would be beneficial for cell tracking algorithms, since cells will be much closer to their previous locations, but may be too harmful to the cells, as discussed above. On the other hand, a sequence of images obtained with a larger time step interval might result in less accuracy in cell tracking, since the cells could have moved further away and even out of focus or might have gone through large morphological changes.

Hence, the aim of this study is to investigate the relationship between the choice of the time step interval and the accuracy of the cell tracking algorithm, within the context of a fully-automated positive feedback loop between the quality of tracking cells and the quality of data acquisition. To the best of the authors’ knowledge, such a study has not been carried out. The majority of cell tracking algorithms published have only considered experimental data as static and given information. Here, we conduct specific experiments with known outcomes (i.e., the direction of cell migration) by either using an effective chemoattractant or employing no-chemoattractant. By acquiring images at 30 s time step intervals, we generate relatively large datasets on which we employ our cell tracking algorithms. To avoid generating extra data at say 1-, 2-min time step intervals, we use the same dataset generated with the 30-s time step interval and simply read out results at other larger time step intervals, to form different subset datasets with different time step intervals. We then evaluate the accuracy of our existing cell tracking algorithms to illustrate the effects of these different time step intervals. We establish that there exists certain fundamental relationships between the tracking accuracy and parameters that are associated with experimental microscope data acquisition.

Our fully-automated approach differs from the current-state-of-the-art cell tracking methodologies in that we investigate the dynamic interplay between segmentation and tracking (which are the hallmarks of the cell tracking algorithms) and the variable adaptive time step interval for generating quality image datasets from the microscopes. We recommend an adaptive approach, where, if the images being generated by the microscope for a particular time step interval are detected by the algorithm to be of less quality (below a certain threshold, hence poor cell tracking), the time step should then be reduced by some factor, automatically (on the fly). Conversely, if the images that are generated by the microscope at a particular time step interval are of high quality (above a certain threshold, hence too high quality cell tracking), then the time step interval can be increased by some factor, automatically. This positive interplay between the quality of the segmentation and cell tracking and the time step interval naturally leads to an adaptive optimal time step selection, such that the quality of the images is appropriate for quality segmentation and tracking and vice-versa.

This research provides a new perspective and benchmark results for the cell tracking community. Multiple experiments involving directed and random cell migration are used to capture live-cell migrations with two different cell lines in order to carry out these investigations. We include both induced (i.e., chemotactically driven) and non-induced experiments that result in directed and random cell migration, respectively. Phase-contrast microscopy is chosen to record the experimental data. This type of microscopy is generally considered to be less harmful to cells, since the intensity of lighting is an order of magnitude lower than other types, such as fluorescence microscopy. Besides the non-destructive light intensity, phase contrast is also less harmful to cells than fluorescence microscopy, because it is a label-free method [29]. We select a number of time step intervals and evaluate each result against expected cell migration pathways and the corresponding manual tracking indices, respectively. We first select two representative algorithms [30] to test automatic cell tracking and then consider a new algorithm that is based on an improved segmentation method.

## 2. Materials and Methods

### 2.1. Cell Culture and Microscopy Data

The experimental datasets used in this study consist of experimental biological data generated in the laboratories of ibidi GmbH. The data demonstrate the migration of two types of adherent, slow-moving cells: human fibrosarcoma cell line HT1080 and normal human epithelial keratinocytes (nHEK). Chemotaxis of slow-moving cells plays a crucial role in various physiological processes, including metastasis and wound healing [31,32,33]. Cell types designated as slow-moving employ the mesenchymal-like migration strategy and their average speed is in the range of approximately one cell-body length per hour (i.e., less than 1 μm/min); in contrast to so-called fast-moving cells, such as immune cells that migrate using the amoeboid type of locomotion at cell speed reaching approximately 10 μm/min [2,34]). The migration speed and the cell-body size provide a cue for selecting a time-lapse interval that is suitable for manual tracking, allowing for the human operator to identify one single migrating cell entity in two consecutive frames. Typically, for slow-moving cells this is of the order of several minutes, e.g., a 5- or 10-min time step interval is generally used when observing the chemotaxis of HT1080 cells [33,35]. The experiments with chemotaxis were performed while using the μ-Slide Chemotaxis assay (ibidi GmbH, Gräfelfing, Germany). Briefly, the HT1080 cells (DSMZ, Braunschweig, Germany) were cultured in Dulbecco’s modified Eagles Medium (DMEM, Sigma, Germany) supplemented with 10% fetal bovine serum (Sigma, Germany). The nHEK cells (CellSystems, Troisdorf, Germany) were cultured in DermaLife basal medium (Lifeline Cell Technology, Frederick, MD, USA) that was supplemented with DermaLife K LifeFactors kit. The cultures were maintained at 37 ${}^{\circ }$C and 5% CO${}_{2}$. After reaching 80% confluence, the cells were trypsinised and a cell suspension of 1–2 ×$10^{6}$ cells/mL was filled into the observation channels of the μ-Slide Chemotaxis, ibiTreat (ibidi GmbH, Gräfelfing, Germany). For experiments with the nHEK cells, the observation channel was beforehand coated with fibronectin. After attachment of the cells, the reservoirs of the chemotaxis chamber were filled with media, as follows:**Dataset** **1:**Both reservoirs and the channel were filled with DMEM supplemented with 10% FBS to induce random migration of HT1080.**Dataset** **2:**The lower reservoir was filled with DMEM with 10% FBS, while the channel and upper reservoir were filled with DMEM without FBS to establish a gradient inducing directed migration of HT1080.**Dataset** **3:**The upper reservoir and the channel were filled with the DermaLife medium with 10 ng/mL epithelial growth factor (EGF; Sigma, Germany) and the lower reservoir was filled with DermaLife medium that was supplemented with a mixture of growth factors, containing 0.5 μg/mL EGF and 50 ng/mL transforming growth factor β-1 (Peprotech, Hamburg, Germany). At these conditions, the gradient of growth factors did not induce a chemotactic response in nHEK cells.**Dataset** **4:**Both reservoirs and the channel were filled with the DermaLife medium with 10 ng/mL EGF to induce random migration of nHEK.

Time-lapse imaging was performed up to 24 h with a time step interval of 30 s, while using a Nikon TiE inverted microscope with a 4× phase-contrast objective (the pixel size = 1.61 μm/px). We exhibit the first frames from each dataset in Figure 1.

### 2.2. Cell Tracking Techniques

In general, cell tracking techniques consist of two stages: segmentation and particle tracking. The first stage is to separate foreground (consisting of objects as well as potential noise and unwanted fragments in motion) and static background. Noise and fragments may be filtered by some characteristic measurements, such as their size. It is important to note that there is some uncertainty that is introduced by noise [12,13,14,15,16,18,19,20,28,30,36,37,38,39,40].

Having obtained a list of objects on every frame from the segmentation, the next stage is to temporally associate corresponding objects between consecutive frames to determine cell movement. Typically, the centroid (i.e., centre of mass) method is used to represent and locate each object (which is treated as a moving particle), again from one image to the other and these are then joined together to form trajectory pathways. Thus, this stage is often termed particle tracking, since objects are reduced to points. Taking a centroid at the current frame as a reference point, if there is a centroid on the following frame and their distance is within a pre-defined range, we say that this movement belongs to the cell that the reference point is representing. A sufficient number of movements from a single cell is recorded as a trajectory and eventually used to analyse its collective migration behaviour. This is the ultimate goal of cell tracking within the scope of this project. The pre-defined range can be seen as a parameter that defines the maximum distance a cell may move within the specified time step interval between two adjacent frames.

Furthermore, it is possible that there are multiple centroids within this pre-defined range for selection. In an ideal situation, one could determine the most appropriate target based on shape changes, similarities in size, cellular dynamics such as head-and-tail identification and statistical analysis to predict movements that are based on previous information. On the other hand, few techniques exist to reliably determine such information. Therefore, we follow arguably the most common approach that assumes the reference point moves to the closest target in that defined vicinity.

In this study, we include two different segmentation techniques following the work by [30]. The first algorithm combines edge detection and active contour [12,38], which uses the edge positions to select the objectives. The second algorithm applies background reconstruction from the entire dataset and subsequently subtract each image from the constructed background image [37,40]. A brief summary of the algorithms is provided below. Further specific details of these techniques can be found in our previous study [30]. In order to demonstrate our approach, we choose the first frame from Dataset 2 described in Section 2.1, which is shown as the top image in Figure 2. On the left-hand side column, we illustrate the first algorithm involving edge detection and active contour. The right-hand column illustrates the application of the second algorithm using the background reconstruction approach. One major issue from the chosen cell lines is that cells often develop filopodia that can be significantly stretched and elongated. This entails that cell shapes can vary drastically. Because the phase-contrast microscopy relies on changes in phase shifts and, depending on the objective lens, it only works in a certain range of phase shifts. If the shift is too big, artefacts are created. Cells may often have a noticeable bright “halo” around their outer rims and their interior may change from dark black to bright white.

These characteristics from phase-contrast microscopy impose specific errors due to the nature of the segmentation techniques. For instance, techniques, such as background reconstruction and subtraction, cannot separate internal intensity changes in a cell from changes around its membrane. For certain elongated cells, cells may be segmented into two fragments, thus causing false positives. On the other hand, an active contour method is robust at detecting changes in the intensity values and may correctly form a connected region to locate cells. Thus, internal differences in a cell can be filtered out and only the interfacial region between the cell membrane and the background is kept. However, an active contour is unlikely to separate cells that are close to each other. In conclusion, background subtraction normally results in more detections (due to fragments) than the ground truth and active contour provides less detections (due to cell proximity). We see these two pattern profiles in Figure 2. The original image has 132 cells in view, the left-hand-side edge detection and active contour found 119 objects, and the right-hand side detects 140.

#### 2.2.1. Segmentation Algorithm 1—Edge Detection and Active Contour

Here we employ some standard segmentation methods, first of all, for each image, we identify the edges by looking at the intensity gradients in that image. This provides the information that is needed for a standard active contour to locate and identify objects of interest. These methods are widely used and their descriptions can be found in [12,38]. We demonstrate these two steps in Figure 2b,c, respectively, and Figure 2d shows the results of using this algorithm.

#### 2.2.2. Segmentation Algorithm 2—Background Reconstruction and Subtraction

The assumption underlying the principles of this method is that we can reconstruct a background image, and then the results of subtraction to the normal images (i.e., containing cells) will yield objects of interest (or noise that may then be removed via various other methods) [30]. To reconstruct the background, we take the mode of the intensity value occurring in all frames at each pixel. Due to the fact that cells are generally moving around (hence the need for tracking), and the density is low, the mode value at each pixel is most likely to be the intensity of the background or a stationary noise that we may also categorise as background.

After having the background reconstructed, all frames can be subtracted from it and, whatever the differences are, these are either moving cells or simply noise. We can then identify and separate cells by their areas and/or trajectories following a common identification procedure [30]. We demonstrate this method in Figure 2e–g, where Figure 2f is the reconstructed background and Figure 2g represents the results after subtraction.

#### 2.2.3. Tracking Algorithm—Nearest Neighbour Approach

We now present our tracking algorithm that relies on results from the segmentation approaches described above. It tracks through locations of centroids and links pairs of centroids from adjacent frames by assessing their distance to a threshold. This has been used in many existing works, for example, in [41,42,43,44]. We further impose a global minimisation to ensure uniqueness of our tracking process. Providing two input datasets from two segmentations, $A(C_{A})$ and $B(C_{B})$, with $N_{A}$ and $N_{B}$ the total number of centroids in datasets *A* and *B*, respectively, and with $C_{A}=\left\{C_{A,i},i=1,\ldots ,N_{A}\right\}$ and $C_{B}=\left\{C_{B,j},j=1,\ldots ,N_{B}\right\}$ the positions of each centroid. We define a threshold distance D>0, such that links between centroids whose distance from each other is greater than *D* is considered to be implausible. Our linking procedure is defined, as follows; first, we compute an $N_{A}\times N_{B}$ array whose *D* entries are given by the Euclidean distance between the *i*th centroid in dataset *A* and the *j*th centroid in dataset *B*, such that
$$d_{i,j}=\left|C_{A,i}-C_{B,j}\right|.$$

We create a link between the centroid $C_{A,i^{*}}$ and $C_{B,j^{*}}$ that minimises the above distance. We then remove the distances that are associated with the links $C_{A,i^{*}}$ and $C_{B,j^{*}}$, i.e., the $i^{*}$th row and $j^{*}$th column of *D*, and apply the above procedure recursively.

## 3. Results

We analyse the four datasets described in Section 2.1 and summarise our findings in this section. The only effective chemoattractant is in Dataset 2, where the chemoattractant is placed at the south side of the view plane. Thus directed cell migration between north and south is expected from this dataset. On the other hand, we expect cells to adopt a random-walk from the other three datasets between north and south directions. Furthermore, all four datasets should show cells adopting a random-walk between east and west directions.

We allow cells to move freely in and out of the boundaries of the view plane; thus, the number of cells in the experiments varies. In addition, a noticeable increase to the cellular area is observed and it is likely due to developed filopodia towards the end of the experiments. We use the confluences [39], i.e., the ratio between areas of cells and total view plane, to evaluate cell density. We summarise the details of each individual experiment below.

**Dataset** **1:**Total number of frames: 2880 (24 h), confluences vary between 7.30–15.02%, random-walk pattern is expected.**Dataset** **2:**Total number of frames: 2880 (24 h), confluences vary between 17.26–48.37%, directional migration towards south, random-walk pattern between east and west is expected.**Dataset** **3:**Total number of frames: 2761 (23.01 h), confluences vary between 4.59–12.44%, random-walk pattern is expected.**Dataset** **4:**Total number of frames: 2566 (21.38 h), confluences vary between 6.42–14.94%, random-walk pattern is expected.

For the first round of tests, we employ all frames (i.e., using a 30 s time step interval) from each individual dataset to track the cells. The second round of tests includes a 1-min interval, meaning we take every second frame in each dataset in our tracking algorithm and completely ignore the intermediates. Thus we only use half of the total number of frames. Tests are further repeated with 2, 3, 5, 7.5, 10, and 15 min intervals. For completeness, the choice of *D* within our tracking algorithm described in the previous section is 10 pixels (roughly around 16.1
μm) for the 30-s interval and gradually increases to around 50 pixels (approximately 83 μm) for the most infrequent time step interval (i.e., 15 min). The minimum threshold on the duration of a cell trajectory is set to be 15% of the experimental duration. In all cases, this duration is roughly 3.6 h, which means a trajectory continually lasting at least 3.6 h is counted as acceptable otherwise it is rejected. There is no upper bound threshold, so a trajectory may theoretically last the full length of the experiment, but this is neither practically common nor possible.

For illustrative purposes, we use the results from edge-detection-active-contour segmentation. We showcase some of the results graphically in Figure 3. We plot all the tracked trajectories of Dataset 2 from four different time step intervals: 30 s, 5, 10, and 15 min, respectively. These trajectories are shown on both white backgrounds (Figure 3(a1–c1)), as well as overlaid on the first image (Figure 3(a2–c2)) of Dataset 2 to give the reader an alternative perspective on the interplay between segmentation and cell tracking. We observe that, as the interval enlarges, less trajectories are found by the cell tracking algorithm. The length of trajectories as well as the accuracy reduce substantially. When considering the standard 10 min setting for manual tracking, we see that this choice of time step interval plays a much more vital role in determining the quality of computerised cell tracking that is different to the manual cell tracking gold-standard approach.

In Table 1, we show all results from the four datasets obtained when using the edge-detection-active-contour segmentation approach. The first column from the left indicates the different choices of the time step intervals. The second column shows the total number of trajectories found, which is equal to or greater than the required duration (i.e., 15% of the total duration of the experiment). The third column shows the number of trajectories whose ending points are on the southern side of the starting reference point. The percentage between south moving and the total number is presented in the fourth column. In the fifth column, we show the number of east moving trajectories and their corresponding percentage in the sixth column. To fully evaluate the performance of our algorithms, we compare these tracked trajectories from our automatic cell tracked results to their corresponding counterpart results obtained manually from a trained human operator. We employ a strict approach, where, if a mistake is made (e.g., tracked to a different cell), we consider such track as a failure to provide accurate information. The details of this comparison are illustrated in the seventh column.

It can be seen from Table 1 that the total number of trajectories decreases as the time step interval gets large. This is demonstrated by all four datasets that confirm our observation, as illustrated in Figure 3. We further observe that the majority of percentages from the random-walk pattern vary around 50±14% with an exception of 29.6%, which comes from a 15 min time step interval with a low identification count. In addition, 87.5% of figures lay between 40.0–60.0%. Whereas, for directed cell migration, all of the figures show a tracking accuracy of more than 70.0%. This result shows that fully-automated algorithms are capable of determining the patterns from experiments of chemotaxis. In terms of high accuracy, only within the 3 min time step interval does the algorithm achieve an accuracy of 90.0% or above.

We further analyse two selected datasets (Dataset 2 and 4) using an alternative segmentation technique that is based on background reconstruction and substitution, as described earlier, in order to demonstrate that patterns are independent of the choice of the technique. It is worth noting that, because of the extra fragments, the tracking accuracy from this segmentation technique is known to be worse. We show the results in Table 2. We can see the accuracy within the 3 min time step interval is only 80% or above. On the other hand, we see that the method is still good enough to distinguish directed cell migration from a random walk. It can be observed that the former stays above 70% in all figures and the latter varies between 40.0–61.5%, with only one exception of 32.3%.

To graphically illustrate the difference between directed migration and random walk, we use the results from two datasets (Dataset 1 and 2). A star-plot format is employed here, which overlays all of the starting points of each trajectory to (0,0) and only links this reference point to the ending point of each trajectory. The results from eight different intervals from these two datasets are shown in Figure 4 and Figure 5.

In terms of efficiency, any test case presented here can be fully-automatically analysed within 40 s on a moderate PC with 3.5 GHZ Intel processors. Whereas, for the human operator, it could take between one to several hours to complete the manual tracking on one data test set.

## 4. Discussion

The interplay between cell tracking algorithms and the time step intervals (the time between when images are taken) associated with image data acquisition techniques has not been explored before in great detail. In this paper, we demonstrated the importance of the time step intervals at the data generation stage and its subsequent influence towards the accuracy of automatic cell-tracking algorithms. The authors are aware of the vast number of cell-tracking research and algorithms, such as the famous Cell Tracking Challenges [14,45] and references within. The majority of current state-of-the-art cell tracking algorithms focus primarily on how best to track cells within existing datasets. In this study, we depart from the current-state-of-the-art and propose a new novel approach, where we investigate optimal time step intervals for quality data acquisition. To demonstrate this concept, we have generated new datasets of variable quality at ibidi GmbH, Germany, where we were able to control the time step interval setting within the microscope. By carrying out manual cell tracking on these datasets, we were able to obtain the gold standard dataset to be used as a measure of accuracy for our novel fully-automated cell-tracking approach. Therefore, we were able to quantitatively investigate the optimal time step interval that yields the best quality of the dataset at a cost-effective manner and with high accuracy in cell tracking. We presented various examples by analysing the data generated from various time step intervals to illustrate the consequences and impact the time step interval has on the quality and accuracy in tracking cells. This approach supports the observation of and argument against exposing cells to too frequent light intensity during experimental data acquisition. With different cell lines, experimental settings, image acquisition devices, and different image analysing methods, an optimal choice of the time step intervals (such as minimising the light exposure while maximising tracking accuracy) is proposed and supported by various results. Therefore, this work opens research avenues and pathways for further interdisciplinary research to improve all relevant areas of cell tracking, which include segmentation, tracking, and adaptive time stepping within microscopes.

## 5. Conclusions

Fully-automated cell tracking has the unrivalled capability of speed for data analytics. In this paper, we establish a relationship between the accuracy in tracking cell trajectories and the time step intervals within which data is acquired that is substantially different from the standard choice of manual tracking. Using four datasets from two cell lines, we presented results which support the theory that our proposed techniques are capable of determining and characterising the migration directions in chemotaxis, thereby providing premises for quantitative statistical analysis. We further argued that, in order to achieve a higher-accuracy in cell tracking from full automation, a much more refined data acquisition approach is needed to compensate for the absence of human intelligence for manual tracking. Our results serve as a good quality control guidance for other experimental data sets.

We recognise that our fully-automated tracking techniques require additional improvement. One aspect of future work is to understand and be able to deal with cell collision. Some recent studies using statistical analysis [44] might provide a good alternative solution. We also believe a better understanding in cell migration is vital and mathematical models from computational fluid dynamics may be a valid tool [46,47].

## 6. Data Management

All of the computational data output is included in the present manuscript. All images that are presented in this paper are available upon request.

## Figures and Tables

**Figure 1 jimaging-06-00066-f001:**
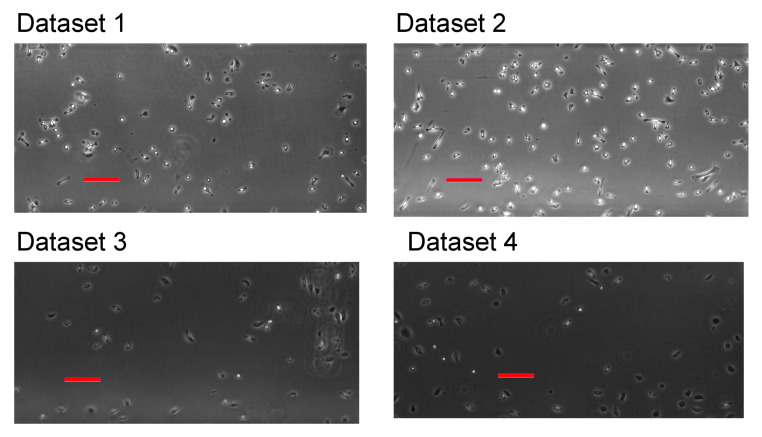
The first frames from each dataset. See main text for description of the datasets. Bars are 200μm in length.

**Figure 2 jimaging-06-00066-f002:**
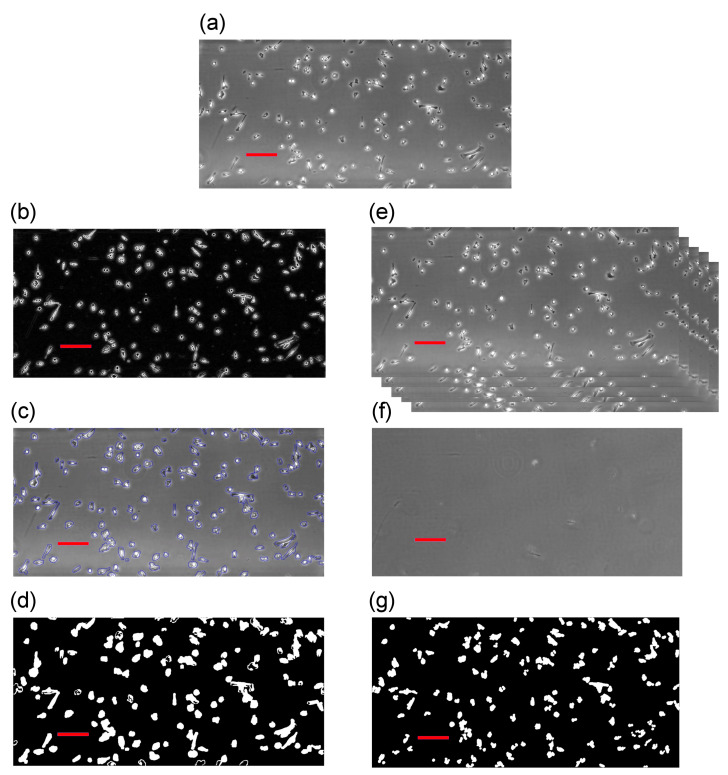
Illustration of two segmentation techniques: (**a**) the original frame from Dataset 1; (**b**) the work of edge detection method [12,38]; (**c**) active contour acting upon the result from edge detection [12,38]; (**d**) binary image as the result from (**b**,**c**); (**e**) illustrating how we stack all images to reconstruct the static background; (**f**) the work of background reconstruction [37,40]; and, (**g**) binary image resulting from the reconstruction and subtraction algorithms [30]. Bars are 200μm in length.

**Figure 3 jimaging-06-00066-f003:**
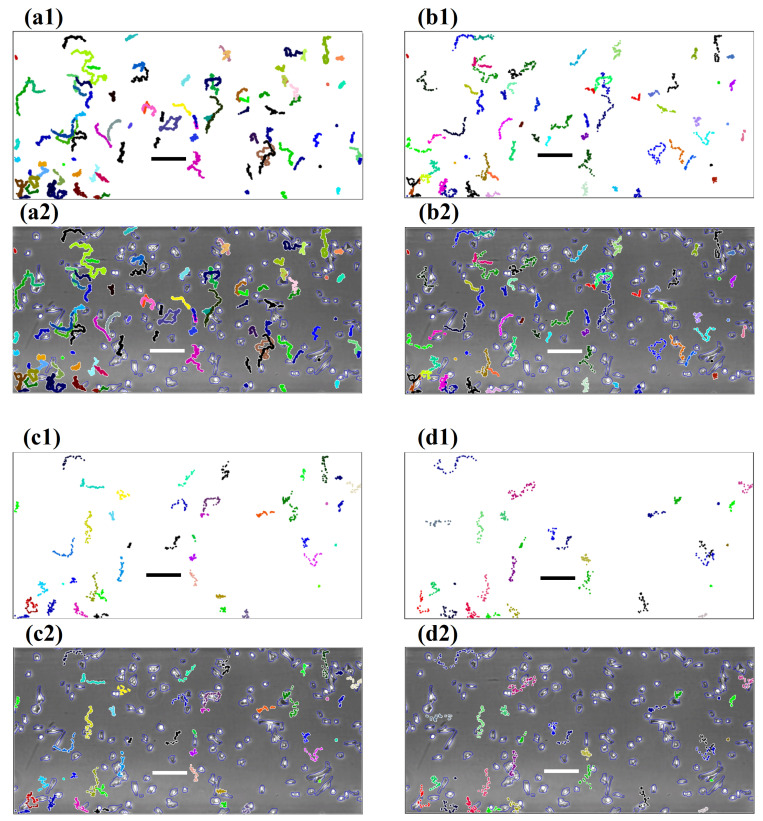
Cell trajectories generated by the fully-automated cell tracking software with different choices of time step intervals. (**a1**–**d1**) These images show trajectories on a plain background. (**a2**–**d2**) Images demonstrate trajectories overlaid on the first image of the sequence. (**a1**,**a2**) Are generated with a 30 s time step interval; (**b1**,**b2**) generated with a 5 min time step interval; (**c1**,**c2**) generated with a 10 min time step interval; and (**d1**,**d2**) generated with a 15 min time step interval. Bars are 200μm in length.

**Figure 4 jimaging-06-00066-f004:**
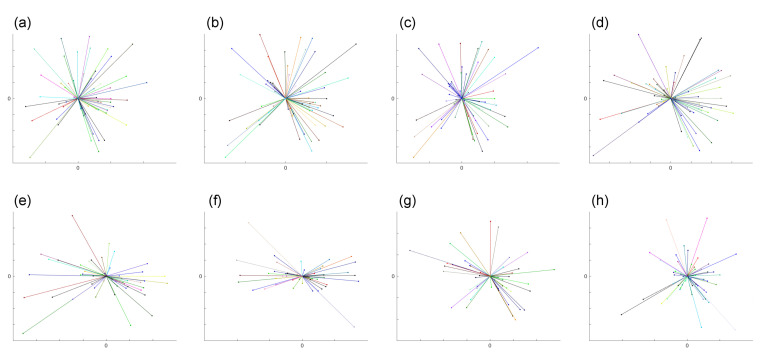
The plots demonstrate graphically the random walk cell migration patterns from Dataset 1 obtained from (**a**–**h**) with time step intervals 30-s, 1-, 2-, 3-, 5-, 7.5-, 10-m and 15-min, respectively. The detailed numbers are shown in Table 1. We overlaid the starting point of each trajectory at the centre point and link the ending point, so as to illustrate the general direction of the cell.

**Figure 5 jimaging-06-00066-f005:**
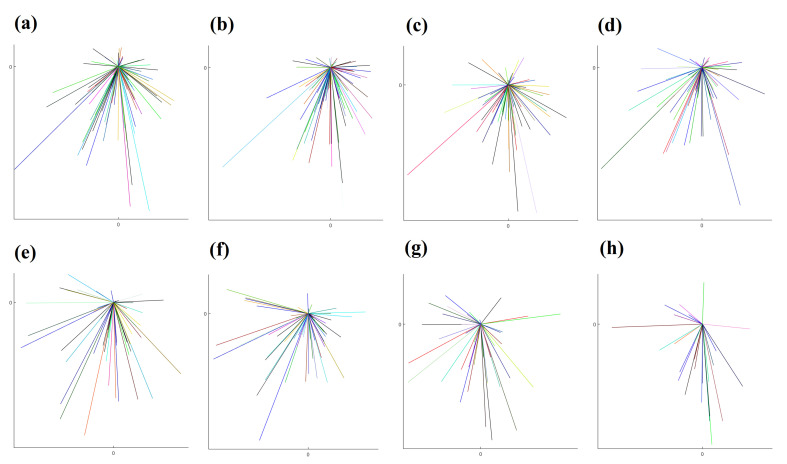
The plots demonstrate graphically directed cell migration patterns from Dataset 2 obtained from (**a**–**h**) with time step intervals 30-s, 1-, 2-, 3-, 5-, 7.5-, 10-, and 15-min, respectively. Table 2 shows the detailed numbers. We overlaid the starting point of each trajectory at the centre point and link the ending point, so as to illustrate the general direction of the cell.

**Table 1 jimaging-06-00066-t001:** Summary of our tracking results from all four datasets using the first algorithm. Accuracy is obtained through comparison between the automatically tracked trajectories to those that have been manually tracked by a human operator.

Time Step Interval	Trajectories	South Moving Trajectories	% of South Moving	East Moving Trajectories	% of East Moving	Accuracy
Dataset 1						
30 s	60	35	58.3%	35	58.3%	98.3%
1 min	58	34	58.6%	32	55.2%	96.6%
2 min	59	33	56.0%	34	57.6%	95.0%
3 min	59	29	49.2%	33	55.9%	93.2%
5 min	51	29	56.9%	30	58.8%	88.2%
7.5 min	53	28	52.8%	31	58.5%	88.7%
10 min	45	21	46.7%	27	60.0%	84.4%
15 min	45	25	55.6%	27	60.0%	84.4%
Dataset 2						
30 s	93	73	78.5%	48	51.6%	96.0%
1 min	87	69	79.3%	42	48.3%	94.2%
2 min	76	59	77.6%	42	55.3%	93.4%
3 min	67	53	79.1%	37	55.2%	93.0%
5 min	56	45	80.4%	29	52.0%	86.0%
7.5 min	63	48	76.2%	32	50.8%	81.0%
10 min	46	35	76.1%	28	60.8%	76.0%
15 min	30	22	73.3%	18	60.0%	66.7%
Dataset 3						
30 s	49	26	53.1%	27	55.1%	93.9%
1 min	49	26	53.1%	26	53.1%	96.0%
2 min	49	25	51.0%	30	61.2%	93.9%
3 min	49	24	49.0%	31	63.3%	92.0%
5 min	44	23	52.3%	26	59.1%	84.1%
7.5 min	40	20	50.0%	24	60.0%	82.5%
10 min	36	19	53.0%	21	58.3%	80.5%
15 min	27	8	29.6%	13	48.1%	74.1%
Dataset 4						
30 s	56	29	51.8%	25	44.6%	91.1%
1 min	57	32	56.1%	26	45.6%	91.2%
2 min	59	33	55.9%	31	52.5%	90.0%
3 min	62	36	58.1%	29	46.8%	88.7%
5 min	55	32	58.2%	27	49.1%	85.5%
7.5 min	51	32	62.7%	20	39.2%	80.4%
10 min	44	24	54.5%	18	40.9%	77.3%
15 min	41	23	56.1%	16	39.0%	73.2%

**Table 2 jimaging-06-00066-t002:** Summary of our tracking results from two selected datasets using the second algorithm.

Time Step Interval	Trajectories	South Moving Trajectories	% of South Moving	East Moving Trajectories	% of East Moving	Accuracy
Dataset 1						
30 s	31	15	48.4%	17	54.8%	89.2%
1 min	36	18	50.0%	21	58.3%	84.4%
2 min	34	18	52.9%	17	50.0%	83.2%
3 min	39	19	48.7%	23	58.9%	81.1%
5 min	21	12	57.1%	12	57.1%	78.8%
7.5 min	18	9	50.0%	11	61.1%	75.5%
10 min	29	17	58.6%	19	65.5%	72.5%
15 min	26	13	50.0%	16	61.5%	67.8%
Dataset 2						
30 s	36	28	77.8%	20	55.6%	83.3%
1 min	38	31	81.6%	21	55.3%	81.6%
2 min	39	31	79.5%	19	48.7%	79.5%
3 min	43	31	72.1%	21	48.8%	74.4%
5 min	41	29	70.7%	20	48.8%	73.2%
7.5 min	39	28	71.8%	20	51.3%	74.4%
10 min	42	31	73.8%	25	59.5%	71.4%
15 min	43	32	74.4%	26	60.5%	65.1%
Dataset 3						
30 s	20	10	50.0%	12	60.0%	93.5%
1 min	19	11	57.8%	10	52.6%	91.1%
2 min	21	11	52.3%	10	47.6%	90.3%
3 min	24	13	54.1%	11	45.8%	85.2%
5 min	22	13	59.0%	12	54.5%	79.0%
7.5 min	21	10	47.6%	12	57.1%	77.2%
10 min	18	8	44.4%	10	55.5%	68.1%
15 min	9	3	33.3%	5	55.5%	55.4%
Dataset 4						
30 s	43	19	44.2%	20	46.5%	86.0%
1 min	42	23	54.8%	22	52.4%	85.7%
2 min	45	26	57.8%	24	53.3%	82.2%
3 min	41	22	53.7%	22	53.7%	80.5%
5 min	44	26	59.1%	22	50.0%	79.5%
7.5 min	39	24	61.5%	19	48.7%	77.0%
10 min	31	18	58.1%	10	32.3%	64.5%
15 min	23	13	56.5%	13	56.5%	43.4%

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
