# Peer review of "Investigating Optimal Time Step Intervals of Imaging for Data Quality through a Novel Fully-Automated Cell Tracking Approach"

_2313-433X, 2020, doi:10.3390/jimaging6070066_

Round 1

Reviewer 1 Report

“A fully automated cell tracking approach for directed cell migration” by Yang, et al. describes tracking cells in time step intervals of 30 seconds to 15 minutes based on established methods. Using the human fibrosarcoma cell line HT1080 and normal human epithelial keratinocytes (nHEK), the researchers show two segmentation techniques and present results tracking cells with eight different time step intervals. The segmentation techniques combine edge detection and active contour, and the other employs background reconstruction. The paper concludes that there is a correlation between the accuracy in tracking cell trajectories and the time step intervals within which data is acquired. The research is presented as support for existing methods in practically tracking cells.

            Several deficiencies render the manuscript unsuitable for publication in its current form. While the paper is a fair assessment of testing existing methods, the paper is written with too many generalizations, which convolute the main point. Also, the paper would be more effective if they tested more variables other than time step intervals.

Several issues should be addressed:

  • The writer should better discuss the algorithms they’re using. While it’s not necessary to reproduce the derivation of the algorithms, a figure or discussion clarifying the technique seems prudent as these are the algorithms the paper is testing.
  • Discuss and present evidence for why the accuracy of the algorithms fall when conditions change.
  • The abstract is unclear. The microscope parameters that are tested should be specified.
  • The line “However, while the majority of these works focus primarily on how best to deal with existing data, with the help from ibidi GmbH, Germany, we are able to control this time step interval setting and using our cell-tracking algorithm as an example, to illustrate the consequences and impact the time step interval would have had on the quality and accuracy in tracking cells, by analyzing the data from various time step intervals.” It’s not evident why working with existing as opposed to newly created data is relevant. Differences between data should be specified and explained. Varying the time-step interval is the point, not an example, of the paper based on line 64, “Hence, the aim of this study is to investigate the relationship between the choice of the time step interval and the accuracy of the tracking algorithm.”
  • It is not clear in Figure 3 the difference between a1 and a2, b1 and b2, etc.
  • The title does not appear to represent the work.

Author Response

Dear Reviewer,

We thank you for your constructive comments. We have addressed your comments in detail as illustrated in the attached report. We hope that our responses are adequate. 

We look forward to your decision.

With best wishes,

Anotida

Reviewer 2 Report

In the paper: “A fully automated cell tracking approach for directed cell migration”, Yang et al. report an automatic cell tracking platform. They claim that the objective of their study is to investigate the relationship between the time interval of imaging and the cell tracking data quality.  I would consider the paper as publishable in this journal, but only after much improvement with major revisions to address a few comments below.

  1. The title of the manuscript is over claimed. The title seems to claim a novel cell tracking approach, but the main claim of the study is the investigation of the relationship between the time interval of imaging and the cell tracking data quality. The main claim or the title should be corrected for appropriateness of the manuscript.

  1. There is a very little explanations in algorithms used in this study and some references in the introduction section are missing. Is there something new in the two algorithms that were used in the study? Why did the authors choose to compare the two algorithms? Is there a quantitative reason for the selection of the demonstrated algorithms?

  1. The algorithms used seem to be bit outdated. Could the authors compare the state-of-art cell tracking algorithms in quantitative manner?

  1. The claim “fully-automated” seems to be not supported by the data presented in the manuscript. Could the authors present more data or explanations regarding full automation?

  1. Figure 4 and figure 5 are not labelled and very little discussion regarding the two data is presented. More discussion in the results displayed in figure 4 and 5 should be elaborated. Also, a quantitative compare chart of the results in these figures should help the readers to understand the main message of these figures.

  1. The main advantage of the developed fully-automated approach compared to other state-of-art technologies is not presented well. This should be explicitly described for the readers. Also, the data is acquired from one institution. Can authors provide a quantitative reason for this? If the manuscript is to claim fully-automated approach that researchers can take advantage of, the authors should perform more experiments with other data produced from different automatic microscopes.

  1. There is no reference to cell migration in the abstract of the manuscript. The abstract should be re-written to fully address the title and the data presented in the manuscript.

  1. In the conclusion section, the authors claim that the results should “serve as a good quality control guidance for other experimental data sets.”

Author Response

Dear Editor-in-Chief and Reviewers,

We thank the Editor-in-Chief and the Reviewers for their excellent support and comments, respectively. We have addressed substantially the comments from the reviewers. Our responses are in the attached report and these are highlighted in blue. We have also uploaded to pdfs, one with highlighted text to aid reviewers to track our changes and another for editorial purposes. 

The comments from the reviewers have helped us to improve substantially the readability and exposition of our findings and hope that it is now appropriate for the journal.

With best wishes,

Anotida

Round 2

Reviewer 1 Report

“Investigating optimal time step intervals of imaging for data quality through a novel fully-automated cell tracking approach” by Yang, et al. describes tracking cells in time step intervals of 30 seconds to 15 minutes based on established methods. Using the human fibrosarcoma cell line HT1080 and normal human epithelial keratinocytes (nHEK), the researchers show two segmentation techniques and present results tracking cells with eight different time step intervals. The segmentation techniques combine edge detection and active contour, and the other employs background reconstruction. The paper concludes that there is a correlation between the accuracy in tracking cell trajectories and the time step intervals within which data is acquired. The research is presented as support for existing methods in practically tracking cells.

            The authors adopted the recommended changes and this revised paper is suitable for publication. The paper is a fair assessment of testing existing methods by varying the time step interval of imaging.

Reviewer 2 Report

The authors have addressed all the points except one. The axes for the figures 4 and 5 still seems to be not labelled. The axes for the trajectory must be defined. If this is addressed I believe the manuscript improved satisfactorily for publication in J. Imaging.